# pDETR: End-to-End Object Detection via Perspective-Aware Transformers

## Abstract

DETR has made notable performance improvements in object detection tasks by leveraging the long-range modeling capabilities of Transformers, but encoding all tokens indiscriminately significantly escalates computational cost and leads to slow convergence. Recent sparsification strategies effectively reduce computational cost through sparse encoders. However, these methods rely heavily on a fixed sparse ratio, which overlooks the coherence of feature representation across levels, leading to performance degradation in complex scenes. To address this issue, we propose a novel object detection approach aimed at constructing consistent representations of multi-level features. The approach composes two steps: First, we introduce a perspective proposal module that leverages the spatial information of high-level foreground features to guide the sparse sampling of low-level features, ensuring both integrity and coherence of multi-scale feature information. Furthermore, we integrated semantic probability to perform hierarchical and dynamic adjustments to the saliency of queries, thereby refining the semantic interaction among foreground queries. Experimental results demonstrate that on the challenging task-specific VisDrone dataset, our pDETR method enhances AP by 1.8% compared to DINO. On the COCO 2017 dataset, the performance improvement of pDETR is even more apparent, achieving a +2.5% increase in AP: under the 1× schedule, pDETR attains an AP of 51.5%, and under the 2× schedule, the AP further increases to 52.0%. Moreover, it exhibits faster convergence, exceeding 40% AP in just 2 training epochs while reducing computational cost by 13% in terms of FLOPs, indicating superior detection capability.

## 1 Introduction

Object detection is one of the most challenging and influential tasks in the field of computer vision, with widespread applications in autonomous driving, video surveillance, and robotic navigation. In recent years, traditional object detection methods based on convolutional neural networks (CNNs)(Cai & Vasconcelos, 2018; Redmon, 2016) have made significant advancements, thanks to the rapid development of deep learning technologies(He et al., 2016). However, these methods still rely on hand-crafted designs(Ren et al., 2015) for preprocessing, candidate region generation, and post-processing steps, which limit their flexibility and scalability. But the introduction of DETR (Detection Transformer)(Carion et al., 2020) eliminates the dependence on manually designed convolutional detectors. As an end-to-end object detection framework, DETR incorporates the Transformer architecture, harnessing its capacity for long-range modeling, leading to substantial improvements in object detection.(Arnab et al., 2021)

Despite DETR's strong performance on large-scale general-purpose datasets like COCO(Lin et al., 2014), its effectiveness is highly dependent on the scale and quality of the training data, and it suffers from slow convergence(Zhu et al., 2020; Hou et al., 2024b). The core issue lies in the conflict between the sparse distribution of foreground objects and the Transformer's uniform processing of all tokens, which results in negative predictions dominating the training process, thereby requiring more samples to achieve effective convergence. The Transformer was initially designed for natural language processing (NLP)(Vaswani, 2017), where language consists of highly abstract sequences of symbols containing rich syntactic and semantic information. It operates without imposing structural bias on the input sequence, treating all tokens uniformly, and captures long-range dependencies between tokens through the self-attention mechanism. However, in images, many pixels are redundant,

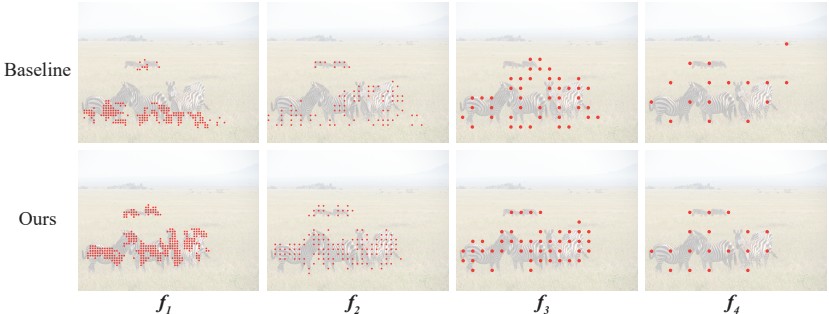

Figure 1: Distribution of foreground tokens retained across feature maps in Baseline and pDETR. Red dots indicate the locations of retained tokens on the original image, adjusted according to the stride. We use Salience DETR as the baseline module to illustrate the semantic misalignment issues.

lacking a specific order or meaning, and applying uniform attention to all tokens results in encoding redundancy. Moreover, flattening two-dimensional images into one-dimensional sequences results in the loss of original spatial relationships(Roh et al., 2021; Zhu et al., 2020). In contrast, convolutional neural networks (CNNs) effectively preserve local spatial information, as their local receptive fields allow the model to focus on neighboring pixel relationships, thereby enhancing feature representation and speeding up convergence. This analysis motivates the introduction of task-specific biases to expedite DETR's convergence and lessen its dependence on large-scale datasets.

Recently proposed sparsification strategies offer new approaches to addressing the redundancy problem caused by the indiscriminate encoding mechanism in Transformers. These methods dynamically select foreground tokens for attention encoding, which not only accelerates the convergence of DETR but also significantly reduces computational overhead. First, Sparse DETR(Roh et al., 2021) proposed selecting the top-$\rho\%$ tokens in the encoder layers using Decoder cross-Attention Map (DAM), thereby reducing the number of tokens processed. Building upon this, Focus DETR(Zheng et al., 2023) introduced ground truth supervision to precisely guide the model in selecting foreground tokens. Salience DETR(Hou et al., 2024a) refines the selection of tokens by introducing a salience mechanism. However, these sparsification strategies rely on foreground score prediction to select foreground tokens, which depend on static sparsification ratios or fixed selection rules. In each feature layer, they choose the tokens with the highest scores for attention encoding, which disrupts information continuity between layers and leads to semantic misalignment. Here, we further conduct a visual analysis of the semantic misalignment issues caused by relying on a static sparsity ratio. As shown in Figure 1, the baseline that depends on a static sparsity ratio exhibits an imbalance in the selection across different feature levels, leading to the omission of some key object tokens.

To address these challenges, we propose the pDETR model, which innovatively integrates a Perspective Proposal Module and a Semantic-Aware Module, aiming to comprehensively enhance the performance of the DETR framework. First, the Perspective Proposal Module uses high-level foreground features to guide sparse sampling in lower layers, maintaining feature transmission across scales and focusing on key features during processing. Secondly, we developed the Semantic-Aware Module, which integrates semantic probabilities to dynamically adjust the saliency of each token at a fine-grained level. This enhances the interaction between semantic information and object queries, thereby improving the model's object detection performance in complex scenes.

The main contributions of this paper can be summarized as follows:

- We propose a Perspective Proposal Module that leverages the spatial position information of high-level foreground features to guide sparse sampling in lower-level features, ensuring the integrity and continuity of multi-scale feature information.

- We design a Semantic-Aware Module that uses semantic probability to perform hierarchical and dynamic adjustments to the saliency of queries, thereby refining the semantic interaction among foreground queries.

- Experimental results on the COCO and VisDrone dataset demonstrate that the proposed method achieves outstanding detection performance while maintaining a low computational cost, significantly improving the efficiency and accuracy of DETR.

## 2 RELATED WORK

**Efficient computation in detection transformer.**

As is well known, DETR suffers from high computational complexity and memory usage(Carion et al., 2020). In efforts to reduce redundant computations, several advancements have been made. Deformable DETR(Zhu et al., 2020) enhances convergence speed by limiting its attention mechanism to a small set of key sampling points around reference points, thus reducing unnecessary calculations. Efficient-DETR(Neubeck & Van Gool, 2006) simplifies the model by initializing object containers with dense priors, which allows for reducing the number of layers in both the encoder and decoder while maintaining detection performance. Lite DETR(Li et al., 2022) introduces an interleaved update mechanism between high-level and low-level features, optimizing the flow of information. PnP-DETR(Wang et al., 2021a) abstracts image feature maps into fine-grained foreground object feature vectors and coarse background context vectors, effectively balancing detail and context. Additionally, recent approaches have adopted sparse encoding strategies to limit the number of queries involved in self-attention within the encoder. Sparse DETR(Roh et al., 2021) refines only the top-$\rho\%$ of tokens across all encoder layers based on Decoder cross-Attention Map results. Focus DETR(Zheng et al., 2023) and Salience DETR(Hou et al., 2024a) introduce token scoring mechanisms within the encoder to prioritize more informative tokens for attention. However, these methods still rely on static sparsity ratios to select foreground tokens, implemented by performing hierarchical top-k selection at each feature level, which fails to fully leverage the complementary nature of features across different layers. Independently performing sparse query encoding at each feature level leads to semantic mismatches and spatial inconsistencies between tokens in different layers. This issue becomes particularly evident when dealing with complex datasets. To address this problem, we propose a method that uses the spatial position information of high-level foreground tokens as priors to guide sparse encoding sampling of lower-level features, ensuring the consistency of semantic and spatial information of the foreground tokens.

**Multi-Scale Features for Object Detection**

In the field of object detection, effectively representing and handling objects at different scales has always been a core challenge. Traditional CNN-based detectors widely employ multi-scale feature extraction techniques such as Feature Pyramid Networks (FPN)(Lin et al., 2017), Bidirectional Feature Pyramid Networks (BiFPN)(Chen et al., 2021), Path Aggregation Networks (PANet)(Wang et al., 2019), and NAS-FPN(Ghiasi et al., 2019). These methods achieve effective multi-scale object detection by fusing features at different scales, significantly improving model efficiency and performance. However, DETR faces significant challenges when processing multi-scale features.

First, handling multi-scale features directly in DETR leads to substantial computational overhead. Since DETR relies on the self-attention mechanism, it processes all tokens indiscriminately, lacking a specialized mechanism to handle objects at different scales(Roh et al., 2021). This global processing not only fails to capture the details of small objects effectively but also wastes considerable computational resources on irrelevant background areas. This issue becomes particularly prominent when processing high-resolution images, where redundant computations are more pronounced. As a result, DETR performs worse in multi-scale scenarios compared to CNN-based detectors, particularly in detecting small objects(Yang et al., 2024).

To address this problem, we propose pDETR. Specifically, we first use spatial information from high-level foreground features to guide the sparse sampling of lower-level features, enabling the model to focus on foreground features while significantly reducing computational costs. Secondly, we utilize semantic probabilities to further refine the focus on foreground areas in a fine-grained manner, preventing distraction by irrelevant background regions. pDETR strikes a balance between accuracy and efficiency, providing a superior solution for multi-scale object detection.

## 3 APPROACH

As shown in Figure 2, pDETR consists of a backbone, a deformable Transformer encoder, and a decoder. The backbone can be equipped with ResNet50 to obtain multi-scale feature maps $\{f_l\}_{l=1}^{L}$ ($L = 4$). Each feature map $f_l \in \mathbb{R}^{C_l \times H_l \times W_l}$, where $C_l$ is the number of channels for each feature map, and $H_l$ and $W_l$ are the height and width of the $l$-th layer feature map. Before being fed into the encoder, the multi-scale feature maps $\{f_l\}_{l=1}^{L}$ ($L = 4$) first pass through the foreground selection module to determine whether each token belongs to the foreground. Then, the selected foreground tokens are passed through the Perspective proposal module, which uses high-level tokens to guide sparse sampling in lower-level features (Section 3.2). These object tokens are then further introduced into the semantic-aware module for fine-grained dynamic adjustment of each token's weight (Section 3.3). Before delving into the specific details, we first review transformers and sparse object detectors.

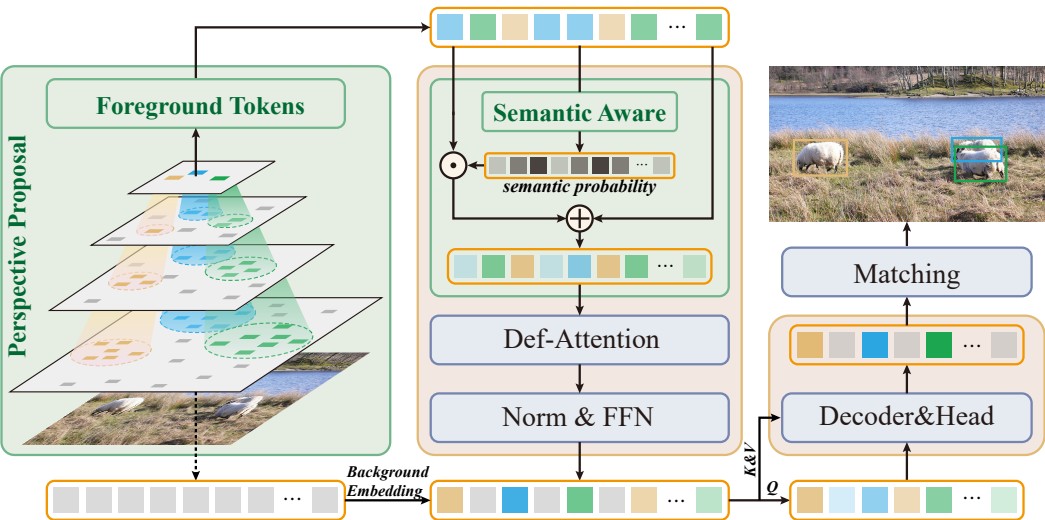

Figure 2: Illustration of the proposed pDETR object detector.

### 3.1 PRELIMINARY

**DETR**

We first review the basic workflow of the DETR(Carion et al., 2020) detector. Initially, it extracts a 2D feature map from the backbone network, denoted as $F \in \mathbb{R}^{H \times W \times D}$, where $H$ and $W$ are the height and width of the feature map, and $D$ is the feature dimension. The feature map is then flattened into a sequence $F_{\text{flat}} \in \mathbb{R}^{N \times D}$, where $N = H \times W$ represents the number of tokens. These tokens are processed by the Transformer encoder, generating enhanced memory $Z \in \mathbb{R}^{N \times D}$. In the decoding stage, DETR introduces a set of learnable object queries $Q \in \mathbb{R}^{M \times D}$, where $M$ denotes the number of queries. The decoder interacts with the output from the encoder and the object queries, predicting a class probability $s_m \in \mathbb{R}^C$ and a bounding box $b_m \in \mathbb{R}^4$ for each query $q_m$, where $C$ is the number of object classes. The final detection results consist of the set $\{(s_m, b_m)\}_{m=1}^{M}$, representing the predictions for the objects in the image.

**Sparse DETR-like Detectors**

To mitigate the high computational cost associated with the self-attention mechanism in DETR, sparse DETR-like detectors introduced sparsification strategies focused on foreground feature selection. These approaches aim to reduce the number of tokens participating in self-attention by selecting only the most relevant foreground tokens, thus improving computational efficiency. **Sparse DETR** employs the Decoder cross-Attention Map from the decoder as a supervision signal, dynamically selecting the top-$\rho\%$ tokens in the encoder layers. Building on this, **Focus DETR** uses a binary supervision signal (foreground or background) derived from ground truth labels, where foreground tokens are marked as 1 and background tokens as 0.

Salience DETR further refines this process by introducing a saliency scoring mechanism. Specifically, Salience DETR assigns a saliency score $\theta^l_{(i,j)}$ to each token $q^l_{(i,j)}$, where the token is mapped back to its position in the original image as $c = (x, y)$. The saliency score is calculated based on the distance $d(c, \text{DBbox})$ between the pixel $c$ and the ground truth bounding box DBbox. Tokens located inside the bounding box receive a non-zero score:

$$\theta^l_{(i,j)} = \begin{cases} d(c, \text{DBbox}), & \text{if } c \in \text{DBbox} \\ 0, & \text{if } c \notin \text{DBbox} \end{cases} \tag{1}$$

where $d(c, \text{DBbox}) = 1 - \sqrt{\left(\frac{\Delta x}{w}\right)^2 + \left(\frac{\Delta y}{h}\right)^2}$, and $(x, y, w, h)$ define the bounding box coordinates and dimensions. Inspired by the Salience DETR method, we assign a saliency score to each token. Specifically, we first determine the high-level foreground tokens through their saliency scores, then use their spatial positions as priors to construct the perspective proposal. In this proposed spatial range, we perform sparse sampling on the lower-level features, thereby avoiding the problem of hierarchical top-k selection disrupting the internal connections between multi-scale features and causing semantic mismatches.

## 3.2 PERSPECTIVE PROPOSAL MODULE

In multi-scale feature representations, high-level features offer rich semantic information crucial for object detection, whereas low-level features capture fine spatial details for accurate localization. To harness the complementary strengths of these hierarchical features, we employ the spatial positions of high-level foreground tokens as priors, progressively constructing a Perspective Proposal to guide sparse sampling in the lower-level features.

In order to capture shape variations more accurately, we introduce a more efficient strategy that replaces the traditional anchor-based approach. By interpolating the positions of high-level foreground tokens onto lower-level feature maps, much like upsampling, we define this resulting mapped area as the Perspective Proposal.

For the query $q^l_{(i,j)}$ in the $l$-th layer, its position $(i, j)$ in the feature map corresponds to coordinate $c = (x, y)$ in the original image, calculated as:

$$c = \left(\left\lfloor \frac{s_l}{2} \right\rfloor + i \cdot s_l, \left\lfloor \frac{s_l}{2} \right\rfloor + j \cdot s_l\right) \tag{2}$$

where $s_l$ denotes the stride in the $l$-th layer, and $i$ and $j$ represent the row and column of the salient token. To map these positions to lower-level features, we use:

$$i' = \left\lfloor \frac{\left\lfloor \frac{s_l}{2} \right\rfloor + i \cdot s_l}{s_{l-1}} \right\rfloor, \quad j' = \left\lfloor \frac{\left\lfloor \frac{s_l}{2} \right\rfloor + j \cdot s_l}{s_{l-1}} \right\rfloor \tag{3}$$

This method efficiently maps salient tokens between different layers, ensuring spatial consistency and allowing salient tokens to propagate across scales. The lower-level salient tokens are selected and encoded based on the positions derived from higher levels. Let $F^{l-1}(q)$ represent the feature value in the $l-1$-th layer, and $P^l$ denote the mapped positions in the $l$-th layer. Salient tokens in the lower layers are selected as:

$$T^{l-1} = \{F^{l-1}(q) \mid q \in P^l\} \tag{4}$$

In the encoder, the salient token sets $T^l$ from all four feature layers participate in self-attention:

$$Q_{T^l} = \text{softmax}\left(\frac{QK^\top}{\sqrt{d}}\right) V \tag{5}$$

where $Q, K, V$ are the query, key, and value matrices, all belonging to $F^l(q)$, and $\sqrt{d}$ is the feature dimension normalization factor.

This layer-by-layer foreground token selection ensures information flow and consistency across feature levels. The Perspective Proposal module enhances the system's sparse query encoding, particularly in complex environments. Importantly, we eliminated isolated foreground tokens, as they lack representational power in lower layers and introduce cumulative errors during interpolation, leading to suboptimal token selection. Various mapping strategies were explored to optimize the Perspective Proposal, detailed in the appendix.

### 3.3 SEMANTIC AWARE MODULE

To enhance semantic interaction among foreground queries, we developed a Semantic Aware Module. This module further refines the localization of target categories within the foreground areas identified by the Perspective Proposal while effectively suppressing background noise. In the DINO model, it was observed that using unrefined content queries as object query initialization could introduce noise that misleads the decoder. Based on this insight and to maintain simplicity in the encoder design, our Semantic Aware Module does not rely on label information but dynamically adjusts the weight of each token in the query using semantic probability information. This approach allows for more precise localization of target categories while effectively suppressing background noise.

The Semantic Aware Module captures the semantic probability information for each salient token across different categories. Specifically, given a set of tokens, each token has predicted scores for $m$ semantic categories. Let $S \in \mathbb{R}^{N \times m}$ represent the matrix of predicted scores for $N$ tokens across $m$ categories. We perform semantic context modeling across four feature layers and generate the semantic context score map $M$ as follows: for each token, the maximum predicted score across all categories is calculated.

$$M = \text{softmax}\left(\max_{i,j \leq C} S_{ij}\right), \quad \forall i \in \{1, \ldots, N\} \tag{6}$$

Here, $M \in \mathbb{R}^N$ contains the maximum predicted score for each token across $C$ categories. The original query embedding $Q_{T^l} \in \mathbb{R}^{N \times d}$, where $d$ is the embedding dimension, is dynamically adjusted using the semantic focus map. The adjusted query embedding $Q'_{T^l}$ is calculated as:

$$Q'_{T^l} = \text{softmax}(M \odot Q_{T^l} + \frac{QK^\top}{\sqrt{d}}V) \tag{7}$$

In this formula, $\odot$ denotes element-wise multiplication. This adjustment allows the model to focus more effectively on tokens with high saliency while reducing interference from background information. Experimental results demonstrate that this module offers significant advantages when handling complex backgrounds or dense scenes, with minimal additional computational cost.

### 3.4 OPTIMIZATION

We design a multi-task loss function for **pDETR** to improve query supervision. The total loss is composed of the standard loss $L_m$, denoising loss $L_{dn}$, focal loss $L_f$, and hybrid loss $L_{\text{hybrid}}$:

$$L_{\text{total}} = \lambda_m L_m + \lambda_{dn} L_{dn} + \lambda_f L_f + \lambda_{\text{hybrid}} L_{\text{hybrid}} \tag{8}$$

Here, $L_m$ includes classification, bounding box regression, and IoU losses, while $L_{dn}$ mitigates noise sensitivity. $L_f$ handles varying object sizes, and $L_{\text{hybrid}}$ supervises both classification and regression. Details are in Appendix A5.

## 4 EXPERIMENT

To comprehensively evaluate the performance of our proposed pDETR's improvements in object detection in large-scale general scenarios and small-scale complex environments, we selected the COCO2017 and VisDrone datasets.

**Dataset** The COCO2017 dataset is a widely adopted benchmark for general object detection in the field of computer vision. In contrast, the VisDrone dataset is specifically designed for object detection tasks from a drone perspective, representing a typical small-scale and complex scene dataset. To provide a clearer comparison between the COCO and VisDrone datasets, we have summarized the key parameters of both datasets, including the number of images and the proportion of large, medium, and small objects, as shown in the table 1:

Table 1: Comparison of COCO2017 and VisDrone Datasets

| Dataset | Images | class | Large Objects | Medium Objects | Small Objects |
|---------|--------|-------|---------------|----------------|---------------|
| COCO2017 | 118,287 | 80 | 264,373 | 219,928 | 160,473 |
| VisDrone | 10,209 | 11 | 133,718 | 445,322 | 1,934,483 |

**Implementation Details**

The implementation details of pDETR are consistent with other detectors similar to DETR. We used the AdamW optimizer to train the model on four NVIDIA RTX 3090 GPUs (24GB each), with weight decay set to $1 \times 10^{-4}$. The initial learning rate was set to $1 \times 10^{-5}$ for the backbone and $1 \times 10^{-4}$ for the other parts of the model, with the learning rate reduced to 0.1 of the initial value during the later stages of training. The batch size per GPU was set to 2, resulting in a total batch size of 4. For the COCO2017 dataset, we conducted 12 and 24 epochs of training, respectively. For the VisDrone dataset, given the smaller object sizes and more complex scenes, we performed 36 epochs of training and adjusted the learning rate at the 30th epoch to further optimize the model's detection performance.

## 4.1 MAIN RESULTS

| Method | Pub/Year | Epochs | Backbone | AP↑ | $AP_{50}$↑ | $AP_{75}$↑ | $AP_S$↑ | $AP_M$↑ | $AP_L$↑ |
|--------|----------|--------|----------|-----|-----------|-----------|---------|---------|---------|
| Deformable-DETR | (Zhu et al., 2020) | 50 | R50 | 46.2 | 65.2 | 50.0 | 28.8 | 49.2 | 61.7 |
| DAB-DETR | (Liu et al., 2022) | 50 | R50 | 48.6 | 66.0 | 50.4 | 29.1 | 49.8 | 62.3 |
| Sparse-DETR | (Roh et al., 2021) | 50 | R50 | 46.3 | 66.0 | 50.1 | 29.0 | 49.5 | 60.8 |
| Anchor-DETR | (Wang et al., 2021b) | 50 | R50 | 42.1 | 63.1 | 44.9 | 22.3 | 46.2 | 60.0 |
| DN-DETR | (Li et al., 2022) | 50 | R50 | 44.1 | 64.4 | 46.7 | 22.9 | 48.0 | 63.4 |
| DINO | (Zhang et al., 2022) | 12 | R50 | 49.0 | 66.6 | 53.5 | 32.0 | 53.2 | 63.0 |
| H-DETR | (Jia et al., 2023) | 12 | R50 | 48.7 | 66.4 | 52.9 | 31.2 | 51.5 | 63.5 |
| Align-DETR | (Cai et al., 2023) | 12 | R50 | 50.2 | 67.8 | 54.4 | 32.9 | 53.3 | 65.0 |
| Focus-DETR | (Zheng et al., 2023) | 12 | R50 | 48.8 | 66.2 | 52.8 | 31.7 | 50.7 | 62.5 |
| Salience-DETR | (Hou et al., 2024a) | 12 | R50 | 49.2 | 67.1 | 53.8 | 32.7 | 53.0 | 63.1 |
| **pDETR** | **Ours** | **12** | **R50** | **51.5** | **68.9** | **56.0** | **35.1** | **55.4** | **67.1** |
| DINO | (Zhang et al., 2022) | 24 | R50 | 50.4 | 68.3 | 54.8 | 33.3 | 53.7 | 64.8 |
| DINO | (Zhang et al., 2022) | 36 | R50 | 50.9 | 69.0 | 55.3 | 34.6 | 54.1 | 64.6 |
| H-DETR | (Jia et al., 2023) | 36 | R50 | 50.0 | 68.3 | 54.4 | 32.9 | 52.7 | 65.3 |
| Align-DETR | (Cai et al., 2023) | 24 | R50 | 51.3 | 68.2 | 56.1 | 35.5 | 55.1 | 65.6 |
| Salience-DETR | (Hou et al., 2024a) | 24 | R50 | 51.2 | 68.9 | 55.7 | 33.9 | 55.5 | 65.6 |
| **pDETR** | **Ours** | **24** | **R50** | **52.0** | **69.8** | **56.2** | **36.0** | **55.6** | **67.1** |

Table 2: Comparison with state-of-the-art methods on COCO val2017 using ResNet50 (IN-1K) as backbone.

To demonstrate the effectiveness of our method, we conducted a thorough comparison with representative state-of-the-art methods on the COCO2017 dataset, as shown in the table 2. pDETR showed significant advantages over both DINO and Salience-DETR. Compared to DINO, pDETR achieved an AP of 51.5% in just 12 epochs, whereas DINO required 24 epochs to reach 51.2%, demonstrating the faster convergence of pDETR. For small object detection, pDETR achieved an $AP_S$ of 35.1%, a substantial improvement over Salience-DETR's 32.7%, highlighting pDETR's stronger capability in recognizing small objects. Similarly, in large object detection, pDETR achieved an $AP_L$ of 67.1%, surpassing Salience-DETR's 63.1%.

| Hyb. | Per. | Sem. | AP | AP$_{50}$ | AP$_{75}$ | AP$_S$ | AP$_M$ | AP$_L$ |
|------|------|------|------|------|------|------|------|------|
|      |      |      | 50.0 | 67.7 | 54.2 | 33.3 | 54.4 | 64.4 |
|      | ✓    |      | 50.5 | 67.9 | 54.8 | 33.8 | 54.3 | 65.5 |
| ✓    |      |      | 50.7 | 68.4 | 54.7 | 33.9 | 54.4 | 65.9 |
| ✓    | ✓    |      | 51.2 | 68.9 | 55.8 | 35.5 | 54.9 | 66.4 |
| ✓    | ✓    | ✓    | 51.5 | 68.9 | 56.0 | 35.1 | 55.4 | 67.1 |

Table 3: Ablation study results on COCO2017.

## 4.2 ABLATION STUDY

The ablation studies on the COCO dataset shown in table3 fully demonstrate the significant improvements brought by our three key enhancements: Hybrid Queries(Hyb.), Perspective Proposal(Per.), and Semantic-Aware(Sem.). First, after introducing Hybrid Queries, the overall AP of the model increased from the baseline of 50.0% to 50.7%, while AP50 and AP75 improved by both 0.7%, respectively, highlighting the contribution of this mechanism in enhancing detection accuracy. Next, with the addition of the Perspective Proposal module, the model's AP further increased to 51.2%, and the small object detection accuracy (AP$_S$) rose from 34.4% to 35.5%, demonstrating the module's effectiveness in capturing small objects in complex scenes. Finally, when we integrated the Semantic-Aware Contrastive Loss module, the model achieved the best performance across all detection metrics, with an overall AP of 51.5%, AP50 reaching 68.9%, and AP$_S$ further improving to 35.1%. In addition, to better illustrate the roles of Perpective Proposal and Semantic-Aware, we visualized the gradient heatmaps under the model's effect, as shown in Figure 3.

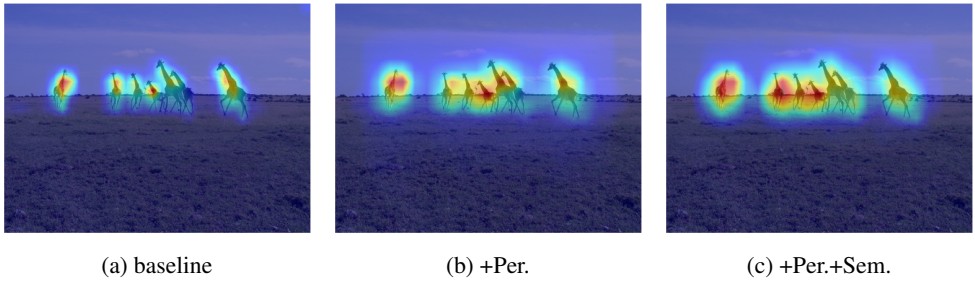

(a) baseline        (b) +Per.        (c) +Per.+Sem.

Figure 3: Comparison of Gradient Heatmaps under Different Modules.

## 4.3 TOWARDS COMPLEX OBJECT DETECTION

| Method | Pub/Year | Backbone | AP↑ | AP$_{50}$ | AP$_{75}$ | AP$_S$ | AP$_M$ | AP$_L$ |
|--------|----------|----------|------|------|------|------|------|------|
| Faster R-CNN | (Ren et al., 2015) | R50 | 21.4 | 40.7 | 19.9 | - | - | - |
| Cascade R-CNN | (Cai & Vasconcelos, 2018) | R50 | 22.6 | 38.8 | 23.2 | - | - | - |
| QueryDet | (Yang et al., 2022) | R50 | 28.3 | 48.1 | 28.8 | - | - | - |
| Sparse-DETR* | (Roh et al., 2021) | R50 | 30.7 | 52.6 | 30.0 | 22.2 | 41.8 | 49.5 |
| CZDet | (Meethal et al., 2023) | R50 | 33.2 | **58.3** | 33.1 | 26.0 | 42.5 | 43.3 |
| Focus-DETR* | (Zheng et al., 2023) | R50 | 31.2 | 52.8 | 30.8 | 22.3 | 42.0 | 52.7 |
| DINO* | (Zhang et al., 2022) | R50 | 32.3 | 54.4 | 32.2 | 23.7 | 43.0 | 52.4 |
| Salience-DETR* | (Hou et al., 2024a) | R50 | 31.6 | 53.3 | 31.8 | 23.1 | 42.4 | 52.8 |
| **pDETR*** | **Ours** | **R50** | **34.1** | 56.5 | **34.3** | **25.2** | **45.2** | **55.6** |

Table 4: Comparison with state-of-the-art methods on Visdrone using ResNet50 (IN-1K) backbone. The * means that we re-implement the methods and report the corresponding results.

The experimental results on the VisDrone dataset, which is shown in table4, not only demonstrate the effectiveness of pDETR in complex scenarios but also reveal the performance differences compared to other models. Interestingly, on the large-scale dataset COCO2017, Salience-DETR significantly outperformed DINO, but on the Visdrone dataset, DINO achieved better results than

Salience-DETR. We believe this is due to the static sparsification ratio used by Salience-DETR, which struggles to adapt well to complex scenes. In contrast, our proposed pDETR achieved the best performance among DETR-like detectors in complex scenarios such as Visdrone, surpassing the strong DINO baseline across all performance metrics, with an improvement of 1.8 AP.

## 4.4 CONVERGENCE ANALYSIS

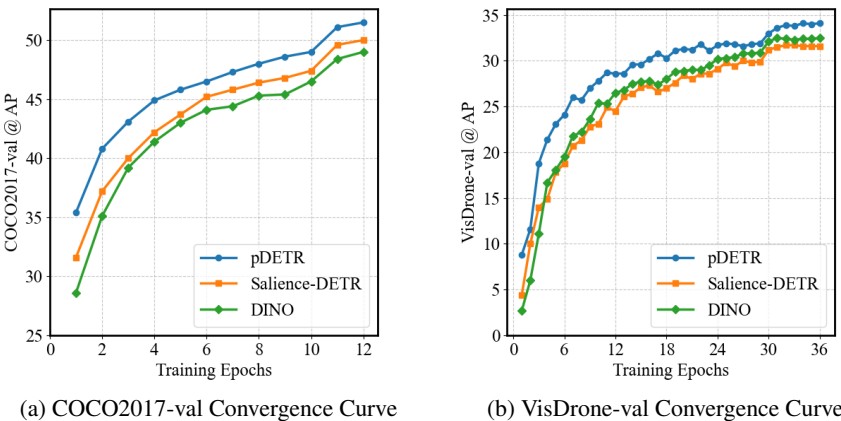

(a) COCO2017-val Convergence Curve      (b) VisDrone-val Convergence Curve

Figure 4: Convergence Curves on COCO2017-val and VisDrone-val Datasets

According to figure 4,the convergence curves on the COCO2017-val and VisDrone-val datasets, pDETR demonstrates a faster convergence rate and higher final performance compared to Salience-DETR and DINO during the training process. In the COCO2017-val dataset, pDETR significantly improved its performance within the first two training epochs, with the average precision (AP) quickly approaching 40%, surpassing both Salience-DETR and DINO. In the VisDrone val dataset, despite the challenges posed by small objects and complex backgrounds, pDETR maintained a rapid convergence speed in the early stages of training. This indicates that pDETR not only excels on large-scale general datasets but also exhibits robust applicability and efficiency when handling complex scenes and small objects.

## 5 CONCLUSION

This paper proposes a novel end-to-end object detection transformer framework, named pDETR. In pDETR, a Perspective Proposal mechanism is introduced to guide sparse sampling in the lower-level features. To enhance semantic interaction among foreground queries, we developed a Semantic Aware Module. This module further refines the localization of target categories within the foreground areas identified by the Perspective Proposal while effectively suppressing background noise. Experimental results demonstrate that our pDETR method achieves significant performance improvements on multiple public datasets. Specifically, it achieves a +2.5% improvement in Average Precision (AP) on the large-scale COCO2017 dataset and a 1.8% AP improvement on the complex-scene VisDrone dataset. Moreover, pDETR significantly accelerates convergence speed while maintaining high-precision detection performance, showcasing an excellent balance between efficiency and performance. We hope that this work will inspire further insights into improving DETR-like detectors.

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

# A APPENDIX

## A.1 IMPLEMENTATION DETAILS OF THE PERSPECTIVE PROPOSAL

The Perspective Proposal Module is designed to resolve spatial and semantic inconsistencies between feature layers by constructing proposals progressively. The process starts with the top feature layer, which contains the richest semantic information, and proceeds downward, ensuring that the higher-layer semantics effectively guide the spatial details in lower layers. For the topmost layer, the original saliency mask is directly used as foreground tokens, determined by saliency scores from Salience DETR (as shown in equation 1). These scores are calculated based on the distance between each token's position in the original image and the ground truth bounding box, ensuring the accuracy of the selected tokens.

In subsequent layers, the foreground token positions from the previous layer are upsampled using nearest-neighbor interpolation to match the resolution of the current layer. This upsampling ensures that the high-level tokens are correctly projected onto the lower-resolution feature maps, maintaining semantic consistency across layers. After upsampling, we refine the foreground token selection by applying a 3x3 convolution to the mask, expanding each token's influence to its neighboring pixels. Tokens with activation values below a set threshold are considered isolated and removed. This process helps ensure that the remaining foreground tokens are semantically relevant and spatially consistent.

These operations are intended to prevent isolated tokens from negatively affecting the next layer's Perspective Proposal. Without removal, these tokens may lead to suboptimal proposals in later layers, as they may lack sufficient information. By iteratively applying this refinement across layers, the Perspective Proposal Module ensures the semantic information from higher layers aligns with the spatial structure in lower layers, improving proposal quality and avoiding the negative effects of isolated tokens.

## A.2 IMPLEMENTATION DETAILS OF THE SEMANTIC-AWARE

The Semantic Aware Module (SCM) dynamically adjusts the query embeddings based on the saliency of tokens across different semantic categories. Given a set of token embeddings $Q \in \mathbb{R}^{N \times d}$, where $N$ is the number of tokens and $d$ is the embedding dimension, the SCM applies a linear classification head that maps the token embeddings to $m$ semantic categories:

$$S = \text{Linear}(Q), \quad S \in \mathbb{R}^{N \times m} \tag{9}$$

Here, $S_{ij}$ represents the predicted score for token $i$ in category $j$. Next, a softmax function is applied to $S$ to normalize the scores across all categories for each token:

$$M = \text{softmax}\left(\max_{i,j \leq C} S_{ij}\right), \quad \forall i \in \{1, \ldots, N\} \tag{10}$$

Finally, the adjusted query embeddings $Q'_{T^l}$ are calculated dynamically by applying the softmax operation and element-wise multiplication over the query matrix in feature layer $T^l$:

$$Q'_{T^l} = \text{softmax}\left(M \odot Q + \frac{Q_{T^l} K_{T^l}^\top}{\sqrt{d}} V_{T^l}\right) \tag{11}$$

In this formula, $\odot$ denotes element-wise multiplication, $Q_{T^l}$ is the query matrix in feature layer $T^l$, and $K_{T^l}$ and $V_{T^l}$ represent the key and value matrices, respectively. $d$ is the feature dimension normalization factor.

This approach allows the model to focus more effectively on the salient tokens, while reducing the interference from background noise.

### A.3 COMPARISON OF DIFFERENT SELECTION STRATEGIES FOR PERSPECTIVE PROPOSAL

We compared two distinct selection strategies for the Perspective Proposal: Layer-by-Layer Perspective and Top-Layer Perspective, to evaluate their performance on the COCO 2017 dataset, as detailed in Table 5. The Layer-by-Layer Perspective strategy progressively transfers high-level feature information across each layer, where the foreground tokens in each layer guide the Perspective Proposal for sparse sampling in the subsequent lower layer. In contrast, the Top-Layer Perspective strategy directly employs the foreground token positions from the highest layer to guide feature selection in all lower layers, bypassing the intermediate layers' transmission process.

| Method | AP $\uparrow$ | $AP_{50}$ | $AP_{75}$ | $AP_S$ | $AP_M$ | $AP_L$ |
|---|---|---|---|---|---|---|
| Layer-by-Layer | **51.5** | **68.9** | **56.0** | **35.1** | **55.4** | **67.1** |
| Top-Layer Perspective | 49.4 | 66.8 | 53.3 | 33.0 | 52.9 | 64.9 |

Table 5: Comparison of methods across different AP metrics.

### A.4 WHY WE NEED SPARSE ENCODING?

| Backbone | Model | epoch | AP |
|---|---|---|---|
| R-50 | DINO | 12 | 48.0 |
| R-50 | Salience DETR | 12 | 50.7 |
| R-50 | pDETR | 12 | 51.5 |

Table 6: Model performance with Hybrid Querie.

| Method | Parameters (M) | FLOPs (G) |
|---|---|---|
| Salience DETR | 56 | 201 |
| DINO | 48 | 298 |
| pDETR (Ours) | 50 | 260 |

Table 7: Comparison of parameters and FLOPs.

We noticed that when using only the highest-level foreground tokens to construct Perspective Proposal, the performance dropped by 2% AP. This is because the highest-level foreground tokens, when mapped to lower-level features, almost covered all regions, failing to focus specifically on the foreground area, and thus could not undergo further refined encoding. It is worth mentioning that we adopted the Hybrid Queries (HQ) strategy. According to the study by H-DETR, directly applying HQ on DINO results in a 1% AP drop. We believe this is because DINO does not perform specific refinement for query tokens. The unrefined tokens add extra supervision signals, which easily lead to erroneous learning in the decoder, thus reducing performance. Similarly, this is also why using only the highest-level foreground tokens to construct Perspective Proposal without further refined encoding leads to a performance drop when using hybrid query supervision.

In contrast, Salience-DETR employed a sparse coding approach with some degree of refinement and achieved a 0.7% AP improvement after introducing HQ. This further demonstrates that sparse coding can effectively reduce interference from irrelevant regions and improve model performance in complex scenarios. However, Salience-DETR still uses a static sparsification ratio, which limits its adaptability to diverse scenes.

Our pDETR adopts a more refined layer-by-layer strategy. It uses Perspective Proposal to sparsely match query tokens with the foreground regions, followed by a semantic-aware refinement step to focus further on foreground objects. Unlike DINO, which processes all query tokens indiscriminately, pDETR's sparse coding approach is more targeted, ensuring that the HQ strategy focuses on useful foreground features. This approach not only avoids interference from irrelevant information but also fully exploits the advantages of sparse coding, leading to better performance across all metrics compared to DINO, while also reducing FLoPS by 13%, significantly improving the model's overall efficiency.

### A.5 THE DETAIL OF LOSS FUNCTION

The focal loss $L_f$ is used to supervise the selection of foreground tokens. The formula for the focal loss is:

$$L_f = -\alpha_t(1 - p_t)^\gamma \log(p_t) \qquad (12)$$

where $\alpha_t$ is a balancing factor used to adjust the weights of positive and negative samples, and $\gamma$ is a modulation factor that controls the model's focus on hard-to-classify samples. The object confidence $p_t$ represents the model's confidence that a sample belongs to the target class, and is calculated as:

$$p_t = \hat{\theta}\hat{\theta} + (1 - \hat{\theta})(1 - \theta) \tag{13}$$

This formula combines the predicted value $\hat{\theta}$ and the ground truth value $\theta$ to generate an assessment of confidence. We keep the same parameter settings as Salience DETR, as $\alpha_t = 0.25$ and $\gamma = 2$.

## A.6 ANALYZING THE COMBINATION OF FOREGROUND AND BACKGROUND

**Foreground + Background Tokens Entering the Decoder:** This strategy allows the decoder to access richer contextual information by incorporating both foreground and background tokens. This approach is particularly effective in complex scenes, as background tokens assist the model in distinguishing between foreground and background objects more accurately. The results indicate that this method shows improved performance in detecting large objects (APL) and medium objects (APM). Specifically, with backbone background encoding, the model achieves an overall Average Precision (AP) of 51.5, AP50 of 68.9, and AP75 of 56.0. The inclusion of additional background information enhances the decoder's ability to grasp the global features of objects. However, it is important to note that this strategy may also introduce unnecessary background noise, which could adversely affect the detection accuracy of small objects (APs), where the model recorded an AP of 35.1.

**Only Foreground Tokens Entering the Decoder:** In contrast, this strategy focuses exclusively on the high-saliency regions of the foreground, minimizing interference from background noise. By processing only salient feature information related to the target, this method excels in small object detection (APs), as it enables the decoder to operate without the distractions posed by irrelevant background information. Without backbone background encoding, the model achieves an AP of 51.2, with an AP50 of 68.9, and AP75 of 55.8; however, the small object detection accuracy (APS) drops to 34.3.

| Backbone | background encoding | epoch | AP | $AP_{50}$ | $AP_{75}$ | $AP_S$ | $AP_M$ | $AP_L$ |
|----------|---------------------|-------|------|-----------|-----------|--------|--------|--------|
| R-50 | ✓ | 12 | 51.5 | 68.9 | 56.0 | 35.1 | 55.4 | 67.1 |
| R-50 | ✗ | 12 | 51.2 | 68.9 | 55.8 | 34.3 | 54.9 | 66.6 |

Table 8: Model performance with and without background embedding.

