# OpenReview forum: "pDETR: End-to-End Object Detection via Perspective-Aware Transformers"
_ICLR.cc/2025/Conference — ICLR 2025 Conference Withdrawn Submission_

### Official Review · Reviewer_CiY8 · 2024-10-27

**Soundness:** 2
**Presentation:** 3
**Contribution:** 2
**Rating:** 3
**Confidence:** 3

**Summary:**

This paper presents a DETR-based object detection framework, named pDETR, which enhances the existing sparse strategies. The approach offers two key contributions. First, it introduces a Perspective Proposal Module that utilizes the spatial positional information of high-level foreground features to guide sparse sampling in lower-level features, thereby ensuring consistency across different feature levels. Second, it incorporates a module that dynamically adjusts the importance of each token based on target categories within the foreground regions. The experimental results demonstrate that pDETR performs better on the COCO val2017 and VisDrone datasets.

**Strengths:**

1. pDETR aims to maintain consistency across feature levels through its perspective proposal module and demonstrates enhanced recognition capabilities, particularly for small objects.
2. pDETR is simple and effective, facilitating ease of implementation and comprehension while surpassing the performance of previous methods.

**Weaknesses:**

1. Unfair Comparison: The baseline presented in Table 3, which does not incorporate the proposed method, already surpasses the Salinece-DETR listed in Table 2. Does this mean that pDETR is trained using different strategies or configurations? Such discrepancies pose an unfair comparison with previous methods and obscure the core contributions of the proposed methods. Besides, FLOPs is missing from Table 2. As an important metric for assessing how many tokens participate in the attention in Encoder, its inclusion is crucial.
2. Insufficient Experiments: The ablation study should include a comparison between the proposed perspective proposal using the positions of current-level foreground tokens and the use of high-level foreground tokens to guide the sampling process. This comparison would more effectively demonstrate the efficacy of the perspective proposal module.
3. Trivial Improvement: As shown in Table 2, pDETR outperforms Salience-DETR by only 0.8 in AP after 24 epochs of training, whereas it achieves a more significant improvement of 2.3 in AP with only 12 epochs of training. This indicates that a shorter training schedule can amplify the observed improvements. So for the ablation results presented in Table 3, the 0.7 gains from combining the perspective proposal module and semantic module may decline with the extended training schedule. These modules may just help the model converge faster but not better.
4. Fact error: Line 399, $AP_s$ of the model equipped with the semantic-aware module declines to 35.1%, rather than improving. This indicates that the semantic module fails to capture the semantics of small objects.
5. Other minor comments and corrections:
Line 65: $F_x$ should be explained in the caption of Figure 1.
Line 123:  Efficient-DETR reference error.
Line 396: 34.4% is not shown in Table 3, it is 33.9%.
Line 457: figure 4 reference format error, it should be Figure 4.
Line 622: A.2 is mostly the same as 3.3, so why not just explain them in 3.3 while there is still room in the main paper?
Line 671: Querie should be Queries.

**Questions:**

Please see weakness section.

---

> ### Author Response · Authors · 2024-11-26
> **Response to Reviewer CiY8's Comments**
>
> Thank you for the detailed review and valuable suggestions regarding our paper. We have carefully reviewed the strengths and weaknesses you highlighted, and we plan to address and improve the following points in the revised version:
>
> 1.Fairness
> We will provide a detailed explanation of the training configurations . Additionally, we will include FLOPs metrics in Table 2 for a more comprehensive evaluation of model performance. It is worth noting that the Salience-DETR results in Table 2 are cited from the CVPR 2024 experimental data, while in our ablation study, we re-evaluated the accuracy of Salience-DETR to ensure consistency and fairness in comparisons.
>
> 2.Experimental Design
> We plan to add ablation studies that compare the use of current-level foreground token positions versus high-level foreground token positions for guiding the sampling process. This will better demonstrate the contribution of the Perspective Proposal Module.
>
> 3.Performance Improvement
> We will further analyze the impact of different training schedules on the observed performance gains in the revised version and discuss whether the contributions of the Perspective Proposal Module and Semantic-Aware Module are primarily reflected in accelerating convergence.

---

### Official Review · Reviewer_mPZ4 · 2024-11-02

**Soundness:** 2
**Presentation:** 2
**Contribution:** 2
**Rating:** 3
**Confidence:** 4

**Summary:**

The paper presents pDETR, an end-to-end object detection model that builds upon DETR by introducing Perspective Proposal and Semantic-Aware Modules. These components aim to address DETR's limitations, such as high computational cost and poor convergence on sparse representations, by selecting and refining critical features at multiple scales. Experimental results show that pDETR achieves improved accuracy and efficiency over state-of-the-art models, particularly on challenging datasets like VisDrone and in detecting small objects.

**Strengths:**

1. pDETR contributes significantly to research on sparse networks by applying sparse encoding techniques within Transformers, providing a notable improvement in reducing computational redundancy for DETR-based models.

2. The method has been tested on two datasets, COCO and VisDrone, demonstrating a degree of generalization across different object detection scenarios and complex backgrounds.

3.  By using the Perspective Proposal and Semantic-Aware modules, pDETR demonstrates superior performance in small object detection, effectively focusing on important features in challenging scenes.

**Weaknesses:**

1. The Method section is somewhat rushed, with certain formula inconsistencies (e.g., in Equation 6 where both "m" and "C" are used to represent categories) that could be simplified. Additionally, the difference between Layer-by-Layer and Top-Layer  Perspective is not clearly explained, leaving readers unclear about how each approach functions.

2. Considering that FLOPs and FPS are critical in sparse network research, the paper does not sufficiently highlight this comparison. Although FLOPs for pDETR are listed in Table 7, a direct comparison with other sparse models is missing.

3.  Figure 2 has large visual elements but lacks clarity in conveying the purpose of Background Embedding, and overall, it contributes little to understanding the method. A more detailed or concise illustration could improve the reader's comprehension of the contributions.

**Questions:**

1. Both approaches are based on position mapping, yet they produce different results. Could you provide a more detailed analysis of their mechanisms, the rationale behind the Layer-by-Layer approach, and how it improves feature consistency through stepwise information transfer?

2. While Salience-DETR applies different sampling rates across layers, pDETR appears to standardize sampling rates between feature maps and encoder layers, yet still achieves better results. Please explain how pDETR balances feature preservation and computational efficiency without varying sampling rates.

3. In Equation 7, are the Q, K, V matrices all token features across layers? If so, would this not cause dimensional mismatches when summing? Additionally, what specific role does $Q^{′}_{T^l}$ play within the encoder, and are similar operations applied to other layers? Further clarification on  $Q^{′}_{T^l}$’s role in encoding would enhance understanding of its functional purpose within the model.

---

> ### Author Response · Authors · 2024-11-26
> **Response to Reviewer mPZ4's Comments**
>
> Thank you for your detailed comments and valuable suggestions. In the revised version, we will address the following issues and provide additional experiments and in-depth analyses:
>
> - Regarding the inconsistencies in the formulas (e.g., Equation 6) and the clarity of the method section, we will refine the descriptions and provide a better explanation of the differences between the Layer-by-Layer Perspective and the Top-Layer Perspective.
> - To address the lack of comparisons with other sparse models (including FLOPs and FPS comparisons), we will include more comparative experiments in the revised version.
> - The illustrations will be redesigned to provide a more detailed explanation of the purpose and implementation of Background Embedding.
> - For the Layer-by-Layer and Top-Layer mapping mechanisms, we will supplement more in-depth experimental analyses in the appendix.
>
> We greatly appreciate the details you have pointed out.

---

### Official Review · Reviewer_swGu · 2024-11-03

**Soundness:** 3
**Presentation:** 2
**Contribution:** 2
**Rating:** 5
**Confidence:** 4

**Summary:**

This paper aims at constructing consistent representations of multi-level features for end-to-end object detection. The authors introduce a perspective proposal module and a semantic aware module to leverage spatial information and semantic probability. Experiments on COCO and VisDrone are shown.

**Strengths:**

- The ablation study shows the effectiveness of proposed modules.
- The experimental results are given on two different datasets COCO and VisDrone.

**Weaknesses:**

- The novelty of proposed method is limited. Both two modules are too straightforward without insightful design.
- The proposed method has limited improvement compared to other methods. Moreover, some related methods, such as DDQ DETR and Relation DERT are not compared. These two methods seem having similar performance as the proposed method.
- This paper only reports the performance using R50. It is necessary to give the experiments using R101 and Swin backbones.
- How about the performance on COCO test set.
- It seems that, compared to the baseline method, the proposed method increased computational cost.

**Questions:**

- The authors can provide more comparisons with the missing methods mentioned in weakness.
- The authors can provide experiments like using different backbones and computational cost.

---

> ### Author Response · Authors · 2024-11-26
> **Response to Reviewer swGu’s Comments**
>
> Thank you for your valuable comments. We have conducted experiments on the suggested larger backbones (e.g., R101 and Swin) and achieved promising results. The experimental findings will be included in the revised version. Additionally, we will supplement comparisons with related methods such as DDQ DETR and Relation DETR, and provide a detailed report on the performance on the COCO test set. Furthermore, we will explore the changes in computational costs and present additional experiments to validate the effectiveness of the proposed method.

---

### Note · Authors · 2024-11-26

**Comment:**

We would like to express our gratitude to the reviewers for their valuable comments and constructive feedback. Upon further reflection, we have realized that the current version of the paper is not sufficiently polished and requires additional improvements to fully address the review comments. We deeply appreciate the time and effort of the reviewers and the opportunity provided by the venue. I have read and agree with the venue's withdrawal policy on behalf of myself and my co-authors.

**Withdrawal Confirmation:**

I have read and agree with the venue's withdrawal policy on behalf of myself and my co-authors.